# Long Chopped Glass Fiber Reinforced Low-Density Unsaturated Polyester Resin under Different Initiation

**DOI:** 10.3390/ma14237307

**Published:** 2021-11-29

**Authors:** Xinjun Fu, Xiaojun Wang, Jinjian Zhu, Minzhuang Chen

**Affiliations:** Department of Composite Materials, College of Materials Science and Engineering, Nanjing Tech University, Nanjing 211800, China; 201961203156@njtech.edu.cn (X.F.); 202061203137@njtech.edu.cn (J.Z.); 202061103111@njtech.edu.cn (M.C.)

**Keywords:** long chopped glass fiber, low-density unsaturated polyester resin, initiation mechanism, radical

## Abstract

Long chopped glass fiber reinforced low-density unsaturated polyester resin (LCGFR-LDUPR) composite materials with light weight and excellent mechanical properties were prepared. It was proved that long chopped glass fiber, which was in length of 15.0 mm and chopped from ER4800-T718 plied yarn, was suitable for the preparation of LCGFR-LDUPR composite samples. With the coexistence of 1.50 parts per hundred of resin (phr) of methyl ethyl ketone peroxide (MEKP-II) and 0.05 phr of cobalt naphthenate, optimal preparation parameters were obtained, which were 20.00 phr of long chopped glass fiber, 2.50 phr of NH_4_HCO_3_, at a curing temperature of 58.0 °C. The lowest dosage of activated radicals produced by MEKP-II and cobalt naphthenate enabled the lower curing exothermic enthalpy and the slowest crosslinking for unsaturated polyester resin to carry out, resulting in a higher curing degree of resin. It was conducive to the formation, diffusion, and distribution of bubbles in uniform size, and also to the constitution of ideal three-dimensional framework of long glass fibers in the cured sample, which resulted in the LCGFR-LDUPR composite sample presenting the apparent density (*ρ*) of 0.68 ± 0.02 g/cm^3^, the compression strength (*P*) of 35.36 ± 0.38 MPa, and the highest specific compressive strength (*P_s_*) of 52.00 ± 0.74 MPa/g·cm^3^. The work carried out an ideal three-dimensional framework of long chopped glass fiber in the reinforcement to low-density unsaturated polyester resin composite samples. It also presented the proper initiator/accelerator system of the lower curing exothermic enthalpy and the slowest crosslinking for unsaturated polyester resin.

## 1. Introduction

Unsaturated polyester resin is one of the most wildly used thermosetting resins due to its low cost and excellent processing performances [1,2,3]. Glass fiber reinforced unsaturated polyester resin composite materials are the most wildly commercial fiber reinforced composite materials for their excellent anti-corrosion and high mechanical properties [4,5,6]. Among them, low-density unsaturated polyester resin (LDUPR) and its composite materials have been well development and drawn attention due to light weight, high mechanical performance and precenting environmentally friendly. It coincides with the development of new lightweight reinforced materials [7,8,9,10].

Chopped glass fiber reinforced composite materials have been a focus of studies in reinforced composite materials of resin matrix. Chopped E-glass fiber and chopped aloe fiber (both in length of 5.0–6.0 mm) were applied in composites of epoxy resin matrix by Jenarthanan et al. [11]. It indicated that more external impact energy absorbed by chopped E-glass fibers and those composite materials presented better mechanical properties. It was reported that E-glass fiber was modified by styrene-butadiene rubber (SBR) and fumed silica, and it was utilized in unsaturated polyester resin to prepare fiber reinforced composites [12]. Below ambient temperature, E-glass fiber reinforced unsaturated polyester resin performed a uniform brittle property, but it exhibited reasonable ductility at and above ambient temperature [13]. Hasan et al. [14] presented composite samples of epoxy resin matrix with filler of expanded polystyrene, chopped E-glass fibers at a length of 3.0 mm or 6.0 mm. It indicated that the chopped glass fiber at a length of 3.0 mm combined well with expanded polyethylene and presented better mechanical properties. Chopped E-glass fiber at a length of 3.0 mm and ammonium bicarbonate (NH_4_HCO_3_) (as the foaming agent) were applied in the preparation of chopped E-glass fiber reinforced low-density unsaturated polyester resin samples by Guo et al. [15]. In their study, a mild thermal mechanism caused by steric retardment of bubbles and fibers was presented; unique microstructure, including “hydrogen bonds”, “bridging” and “spatial obstruction”, was also proposed. All of those studies presented above were conducted in the enhancement of chopped fibers to resin matrix, but the length of chopped fibers was not longer than 6.0 mm.

As for low-density unsaturated polyester resin (LDUPR), novel foaming methods and theoretical models have been highlights of present research. Those methods and models were established in the presence of a single initiator [15,16,17,18,19].

In commercial application, direct yarn and plied yarn were common categories for glass fibers, which were defined according to the process and the treatment of production of glass fiber. Direct yarn, which was single-stranded roving, was a kind of continuous fiber obtained from large bushing with thousands of nozzles. On the other hand, two or more single yarn twisted together through a winder to fabricate plied yarn [20,21,22]. Different sizing agents lead to different softness of glass fibers. Direct yarn is soft. Besides, there are two kinds of plied yarn, which are stiff plied yarn and soft plied yarn. Direct yarn and soft plied yarn are appropriate for filament winding, pultrusion, fabric, etc. Stiff plied yarn is appropriate for spray-up and chopped strand mat. Accordingly, scientific issues, such as the category of glass fiber, reinforcement of different lengths of chopped glass fibers, different initiations for unsaturated polyester resin cross-linking, are novel explorations of resin matrix composite materials. Those are also active research areas in chopped glass fiber reinforced low-density unsaturated polyester resin.

In this study, long chopped glass fiber was firstly applied to reinforce low-density unsaturated polyester resin to prepare long chopped glass fiber reinforced low-density unsaturated polyester resin (LCGFR-LDUPR). Different initiations were explored in order to obtain the optimal preparation parameters, the proper initiation system, and excellent mechanical properties. Effects of chopped glass fibers in different types and in different lengths on the distribution of chopped glass fiber in resin glue were observed and compared in order to select proper type and proper length of chopped glass fiber. Gel time experiment was necessary to determine the proper ratio and corresponding content for initiator and accelerator. Optimal preparation parameters, the proper initiation system, and excellent mechanical properties were obtained through orthogonal experiments under different initiation. Effects of long chopped glass fiber and different initiators on the curing of resin and different initiation mechanisms were explored. The framework of long chopped glass fibers in cured resin matrix and their excellent reinforcement to LCGFR-LDUPR were analyzed by microstructural observation. The work carried out an ideal three-dimensional framework of long chopped glass fiber in the reinforcement to low-density unsaturated polyester resin composite samples. It also presented the proper initiator/accelerator system of the lower curing exothermic enthalpy and the slowest crosslinking for unsaturated polyester resin.

## 2. Materials and Methods

### 2.1. Materials

Unsaturated polyester resin used in the study was orthophthalic polyester resin (Type of P17-902) with a solid content of 63–66 wt%. It was produced by AOC Aliancys Resins Co., Ltd., Nanjing, China. Its viscosity was 1300–1500 mPa·s (at 23 ± 1 °C), and its acid value was 15–19 KOH/g^−1^.

Two kinds of direct yarns, which were ECR13-300D-608 and ECR15-600D-608, were applied in the research; in which ECR represented E-glass of chemical resistance, and D represented direct yarn. Numeral 13 or numeral 15 was the monofilament diameter (μm) of different direct yarns; numeral 300 or numeral 600 meant the linear density (the unit of linear density generally used is tex, which corresponds to 1 g per kilometer of yarn, according to the standard of IOS 1889:2009) of different direct yarns; 608 was the type of sizing agent for direct yarn treating. They were products of Taishan Glass Fiber Co., Ltd., Tai’an, China.

Two kinds of plied yarns were applied in the research, which were ER2400-T628 and ER4800-T718. ER represented E-glass of roving, numeral 2400 or numeral 4800 meant the linear density (tex) of different plied yarn, T628 or T718 was the type of sizing agent for plied yarn treating. The monofilament diameter was 13 μm for ER2400-T628 and for ER4800-T718. They were manufactured by Changzhou Tianma Group Co., Ltd., Changzhou, China. Direct yarns or plied yarns were chopped in lengths of 9.0 mm, 12.0 mm, 15.0 mm, 18.0 mm, and 21.0 mm, then they were added into unsaturated polyester resin glue as reinforcements to prepare long chopped glass fiber reinforced low-density unsaturated polyester resin (LCGFR-LDUPR) samples.

NH_4_HCO_3_, as a foaming agent in the study, was from Shanghai No. 4 Reagent & H.V. Chem. Co., Ltd., Shanghai, China, with a NH_4_HCO_3_ content more than 99 wt%.

Three kinds of initiators were utilized in this study. Among them, methyl ethyl ketone peroxide (MEKP-II) was produced by Luoyang Shuangyue Curing Agent Co., Ltd., Luoyang, China, and with peroxide content of more than 33 wt%. Cyclohexanone peroxide (CYHP) with peroxide content over than 50 wt%, and tert-butyl peroxy-2-ethylhexanoate (TBPO) with a peroxide content ≥98 wt%. CYHP and TBPO were both produced by Akzo Nobel (Tianjin) Co., Ltd., Tianjin, China.

Cobalt naphthenate (NL-49P), which was an accelerator of polymerization of unsaturated polyester resin, was made by Akzo Nobel (Tianjin) Co., Ltd., Tianjin, China, with cobalt content between 7.8 wt% and 8.2 wt%.

PMR-EZ, as a release agent in the study, was obtained from Chem-Trend Chemicals (Shanghai) Co., Ltd., Shanghai, China.

### 2.2. Methods

#### 2.2.1. Microscopic Analysis

Distributions of bubbles and long chopped glass fibers in unsaturated polyester resin glue or in cured composite samples were observed by auto-fine-tune portable microscope (B011, made by Shenzhen Supereyes Technology Co., Ltd., Shenzhen, China).

#### 2.2.2. Viscosity Testing

According to the standard of ISO 2555: 2018, “Plastics–Polymers/resins in the liquid state or as emulsions or dispersions—Determination of apparent viscosity using a single cylinder type rotational viscometer method”, viscosity of resin glue was detected by digital rotatory viscometer (NDJ-79A, made by Shanghai Changji Geological Instrument Co., Ltd., Shanghai, China). The sensor accuracy of NDJ-79A was 2% full scales, and its working relative humidity was required at no more than 80%.

#### 2.2.3. Gel Time Determination

According to the standard of ISO 2535:2001,“Plastics–Unsaturated-polyester resins”, the gel time of unsaturated polyester resin was measured by a resin reaction behavior analyzer (Gelprof 518, Wuhan Jiudi Composite Material Co., Ltd., Wuhan, China), the temperature accuracy of the instrument was 0.1 °C, and its working temperature was below 300.0 °C.

#### 2.2.4. Preparation of LCGF-LDUPR Specimens

According to the standard of ISO 3672-2:2000, long chopped glass fiber reinforced low-density unsaturated polyester resin (LCGFR-LDUPR) samples were prepared according to the normal formula of: 100 resin: W NH_4_HCO_3_: X initiator: Y accelerator: Z chopped glass fiber at a fabrication temperature T °C [23]. Among them, the contents of X, Y, Z, T were determined by corresponding experiments, and the content of the foaming agent was usually controlled within 3.00 phr (parts per hundred of resin) [16,24]. Firstly, certain contents of ground foaming agent, initiator, accelerator, and long chopped glass fiber were added into the unsaturated polyester resin glue. The mixture was then stirred by a stirrer (J J-1A, Changzhou Jintan City Hengfeng Instrument Manufacturing Co., Ltd., Changzhou, China) for 5.0 min at a speed of 120 r/min until long chopped glass fibers distributed homogeneously in resin glue. Later, the mixture was added into a mold uniformly, then cured at a certain temperature for 2.0 h. Finally, samples were cooled down to room temperature and demolded. In a parallel experiment, five samples were prepared for each formulation. The controlling accuracy of the temperature was less than 0.2 °C.

#### 2.2.5. Testing of Apparent Density

The apparent density of the LCGFR-LDUPR sample was detected according to the standard of ISO 845:2006. Each cured sample was cut into a regular cylinder in a diameter of 60 ± 1 mm and height of 50 ± 1 mm. Five samples were tested for each formula, and the average value of apparent density of samples was calculated and obtained.

#### 2.2.6. Testing of Compressive Strength

According to the standard of ISO 844:2014, “Rigid cellular plastics—Determination of compression properties”, each cured sample was cut into a regular cylinder in a diameter of 60 ± 1 mm and 50 ± 1 mm in height; the compressive strength (*P*) of cylinder samples was tested by using an electronic universal testing instrument (WDW3100, Changchun Kexin Testing Instrument Co., Ltd., Changchun, China); the accuracy of the instrument was 0.5% and its maximum pressure was 100 kN. The ambient temperature of the experiment was 23 ± 2 °C and the relative humidity was 50 ± 5%. Five samples were tested for each formula, and the average value of *P* of samples was calculated and obtained.

#### 2.2.7. Thermal Analyses

The curing process of the samples was studied by using differential scanning calorimetry (DSC), which was NETZSCH DSC 204 designed by NETZSCH Scientific Instruments Trading Co., Ltd., Selb, Germany. The modulation temperature of the DSC was 0.1 °C, and sensitivity of the DSC was 0.1 μg. The sample at a weight of 5–10 mg was sealed in an aluminum crucible and in an atmosphere of nitrogen. The nitrogen flow was 30 mL·min^−1^.

##### Non-Isothermal DSC

A non-isothermal scan was performed at a heating rate of 10 °C·min^−1^ in a temperature range from 20 °C to 200 °C. Non-isothermal changes of the curing process of the samples were recorded and analyzed.

##### Isothermal DSC (Testing of Curing Degree of LCGFR-LDUPR Composite Samples)

The sample was analyzed by isothermal differential scanning calorimetry (DSC) at a constant temperature for an hour to obtain the polymerization heat of unsaturated polyester resin which was used as *Q_P_* [25]. Then, the sample was cooled to 20 °C at a cooling rate of 20 °C·min^−1^. Later, the temperature was raised again from 20 °C to 200 °C at a rate of 10 °C·min^−1^ to detect the residual heat (*Q_R_*) of the unsaturated polyester resin [25]. The total heat *Q_T_* of the curing process was calculated by Equation (1). The curing degree α was calculated by Equation (2) [25].
*Q_T_* = *Q_P_* + *Q_R_*(1)
*α* = (*Q_P_*/*Q_T_*) × 100%(2)

## 3. Results and Discussion

### 3.1. Distribution of Long Chopped Glass Fibers in Resin Glue

Four kinds of chopped glass fibers obtained from four types of glass fiber, which were in length of 9.0 mm and with the presence of 10.00 phr, were added and distributed in resin glue, respectively. Distributions of chopped glass fibers in resin glue were observed by auto-fine-tune portable microscope (B011), which are shown in Figure 1.

Figure 1a,c,d indicate that there is no cluster of chopped glass fibers and no deformation of bubbles in the resin glue, including in type ECR13-300D-608, in type ER2400-T628, and in type ER4800-T718. However, in the same length and in the same content, the cluster and tangle emerge for chopped glass fibers obtained from ECR15-600D-608, which is shown in Figure 1b. Moreover, bubbles are enwrapped by chopped glass fibers, and are deformed, resulting in inhomogeneous distribution in resin glue. These phenomena were adverse to mechanical properties of composite samples. It is deduced that under conditions of the same length and the same content of chopped glass fiber, the higher linear density of direct yarn was, the more monofilaments there were in each bundle of glass fiber. More glass fiber monofilaments were easy to cluster, and even to tangle in resin glue. On the other hand, stiff chopped glass fibers acquired from plied yarns easily distributed and avoided cluster and tangle in resin glue due to their better stiffness. Consequently, three kinds of chopped glass fibers obtained from direct yarn in type ECR13-300D-608, plied yarn in type ER2400-T628, and plied yarn in type ER4800-T718 were determined to be used in follow experiments. Distributions of those three kinds of chopped glass fibers in resin glue, which were in the presence of 15.00 phr and in length of 9.0 mm, 12.0 mm, 15.0 mm, 18.0 mm, and 21.0 mm, were observed by auto-fine-tune portable microscope (B011) and compared, and are shown in Figure 2.

As shown in Figure 2a–c,f–h,k–m, there are no tangles and no clusters emerging for chopped glass fibers in lengths of 9.0 mm, 12.0 mm, and 15.0 mm. Meanwhile, there are no deformed bubbles in resin glue because of no tangle or no cluster of chopped glass fibers.

However, as for chopped glass fibers obtained from ECR13-300D-608, ER2400-T628, and ER4800-T718, it is revealed that chopped glass fibers with the presence of 15.00 phr and at a length of 18.0 mm or a length of 21.0 mm gather together to cluster and tangle in resin glue, as shown in Figure 2d,e,i,j,n,o. Furthermore, bubbles are enwrapped by long chopped glass fiber and deformed, resulting in inhomogeneous distribution in resin glue. Those phenomena are obvious for chopped glass fibers at a length of 21.0 mm, as shown in Figure 2e,j,o. Usually, it was considered that the longer length of chopped glass fibers was, the higher mechanical properties of the composite materials were [26,27]. According to the viewpoint, longer length of chopped glass fibers was favorable to enhance mechanical properties of long chopped glass fiber reinforced low-density unsaturated polyester resin (LCGFR-LDUPR) samples. Therefore, 15.0 mm was considered to be the suitable length of chopped glass fibers in the preparation of LCGFR-LDUPR samples, according to distribution experiments of long chopped glass fiber in resin glue.

### 3.2. Effects of Glass Fiber Types, the Content of Chopped Glass Fiber on the Viscosity of Resin Glue

It was pointed out that the movement of the polyester molecule chains was hindered as the steric obstruction increased in the resin glue, and the viscosity of the resin glue increased with the addition of the enhancement [28]. Therefore, viscosity changes of unsaturated polyester resin glue were affected by the length and the content of the chopped glass fiber, even influenced by the type of glass fiber.

Effects of different contents of chopped glass, such as 10.00 phr, 15.00 phr, 20.00 phr, 25.00 phr, and 30.00 phr, on changes of viscosity were detected and are shown in Figure 3. All of those chopped glass fibers were obtained from three types of glass fiber which were ECR13-300D-608, ER2400-T628, and ER4800-T718. In the case of a certain length of chopped glass fibers, Figure 3 illustrates that the viscosity of resin glue increases with the increase of the content of chopped glass fiber in resin glue. Consequently, it is deduced that as the increase of the content of long chopped glass fibers, steric hindrance of long chopped glass fibers suppressed the movement of polyester molecular in resin glue, and the viscosity of resin glue increased as well as with the developing steric retardance increase.

Figure 3 illustrates that under the same addition of chopped glass fibers got from direct yarn of ECR13-300D-608, the viscosity of resin glue was much higher than the other two kinds of resin glue with chopped glass fiber got from different plied yarns. It is because the monofilament of direct yarn was different from that of plied yarn. The monofilament of direct yarn was soft, but the one of plied yarn was stiff. In spite of the account of monofilament of plied yarn being much more than that of direct yarn, which was 2400 monofilaments for plied yarn and 300 monofilaments for direct yarn, soft monofilament of direct yarn was easy for winding and twisting each other in resin glue. Therefore, steric retardance to the movement of linear polyester molecules was evident for the presence of chopped glass fibers got from direct yarn of ECR13-300D-608, resulting in an increase of viscosity of resin glue. It is deduced that stiff plied yarns were favorable for the preparation of long chopped glass fiber reinforced low-density unsaturated polyester resin (LCGFR-LDUPR) composite samples.

As for plied yarns, changes of viscosity of resin glue with the presence of long chopped glass fibers got from ER2400-T628 glass fiber are a little lower than those with the presence of long chopped glass fibers got from ER4800-T718 glass fiber, which are shown in Figure 3. It is concluded that under the condition of a certain content and a certain length of chopped glass fibers, higher linear density represented more monofilaments in each bundle for plied yarn. Therefore, more monofilaments meant more steric retardance in resin glue, resulting in the increase of the viscosity of resin glue.

Distributions of long chopped glass fibers at a length of 15.0 mm and in five different contents, and obtained from two types of plied yarn (ER2400-T628 and ER4800-T718), were observed by auto-fine-tune portable microscope (B011), and are illustrated in Figure 4. Figure 4a–c,f–h show that long chopped glass fibers, which are in the content of no more than 20.00 phr, distribute homogeneously in resin glue. As the content of long chopped glass fibers (got from ER2400-T628 glass fibers) reaches up to 25.00 phr, cluster and tangle of long chopped glass fibers emerge in the resin glue and are shown in Figure 4d. It is adverse to the preparation and properties of LCGFR-LDUPR composite samples. However, long chopped glass fibers (got from ER4800-T718 glass fibers) distribute homogeneously in resin glue in spite of the content of 25.00 phr (see Figure 4i). It is considered that in the case of a certain content and a certain length of chopped glass fibers, glass fiber bundles for plied yarn of 2400 tex were more than those of 4800 tex. The more glass fiber bundles were owned for plied yarn, the more difficult and inhomogeneous was the distribution for glass fiber bundles in resin glue. As a result, there is cluster and tangle in the resin glue for 25.00 phr of long chopped glass fibers obtained from the ER2400-T628 glass fiber.

As for 30.00 phr of long chopped glass fibers obtained from ER2400-T628, cluster and tangle emerged in the resin glue, and are shown in Figure 4e. Those phenomena also emerged for 30.00 phr of long chopped glass fibers obtained from ER4800-T628, shown in Figure 4j. Usually, the more content of chopped glass fibers, the higher mechanical enhancement for chopped glass fibers reinforced composite samples [29]. Accordingly, the content of this kind of long chopped glass fibers at a length of 15.0 mm was controlled within 30.00 phr in following experiments of preparation, thermal analyses, and microstructure analyses.

### 3.3. Coordination of Methyl Ethyl Ketone Peroxide and Cobalt Naphthenate

In commercial application, the gel time of resin glue is usually controlled between 23.0 min and 35.0 min [25,30,31]. According to previous research, the foaming agent NH_4_HCO_3_ decomposed in a temperature range from 50.0 °C to 58.0 °C [15,17]. Therefore, coordination between methyl ethyl ketone peroxide (MEKP-II) and cobalt naphthenate was proposed to match the commercial gel time of resin glue.

In the coordination experiment between MEKP-II and cobalt naphthenate, the curing temperature was set at 50.0 °C or at 58.0 °C, and the content of MEKP-II was set in 2.00 phr. It could avoid the influences of initiator, accelerator, and curing temperature, and could conveniently determine the proper ratio of MEKP-II and cobalt naphthenate. According to this experimental design, changes of gel time of resin glue were detected under variant contents of cobalt naphthenate to 2.00 phr MEKP-II at a temperature of 50.0 °C and at a temperature of 58.0 °C, respectively. Corresponding changing results of gel time of resin glue are listed in Table 1.

Table 1 shows that with the ratio of MEKP-II to cobalt naphthenate being 30:1 (2.00 phr of initiator to 0.07 phr of accelerator), the gel time of the resin glue is 31.5 ± 0.6 min at 50.0 °C, and 21.0 ± 0.3 min at 58.0 °C. Those results of the gel time were close to the practical one, which is from 23.0 min to 35.0 min. Therefore, the proper ratio of MEKP-II to cobalt naphthenate was set 30 to 1 in a temperature range of 50.0–58.0 °C.

Based on the above discussion, in the temperature range of 50.0–58.0 °C, the ratio of MEKP-II to cobalt naphthenate was 30 to 1, and changes of gel time with varied contents of MEKP-II are shown in Figure 5. Figure 5 illustrates that with the presence of 1.50 phr MEKP-II (corresponding to 0.05 phr of cobalt naphthenate), the gel time of resin glue is between 23.2 ± 0.2 min and 34.6 ± 0.4 min, which was suitable for commercial application. Therefore, in the preparation of LCGFR-LDUPR, the proper addition of MEKP-II was 1.50 phr together with 0.05 phr of cobalt naphthenate.

### 3.4. Preparation of LCGFR-LDUPR Composite Samples

According to the above range of curing temperature, the ratio of methyl ethyl ketone peroxide (MEKP-II) to cobalt naphthenate was 1.50 phr to 0. 05 phr, and the length of chopped glass fibers was 15.0 mm. Three factors, such as curing temperature, the content of foaming agent NH_4_HCO_3_, and the content of long chopped glass fiber at a length of 15.0 mm were selected as critical factors. Corresponding experimental data for those three factors were decided as follows. The curing temperature range was from 50.0 °C to 58.0 °C in intervals of 2.0 °C, the addition of NH_4_HCO_3_ foaming agent was from 1.00 phr to 3.00 phr in intervals of 0.50 phr, and the addition of long chopped glass fiber (in length of 15.0 mm) was from 10.00 phr to 30.00 phr in intervals of 5.00 phr. Subsequently, an orthogonal experiment of L_25_(5^3^) (where L is the code name, numeral 3 represents the number of factors, numeral 5 is levels of each factor, and numeral 25 represents serial number of samples) was designed to explore optimal parameters for preparation of long chopped glass fiber reinforced low-density unsaturated polyester resin (LCGFR-LDUPR) composite samples. Results of orthogonal experiment are listed in Table 2.

Under a level of a certain factor, the mean *k* and the range *R* of apparent density (*ρ*), compressive strength (*P*), and specific compressive strength (*Ps*) of LCGFR-LDUPR composite samples were obtained through calculation, and analysis of experimental results are listed in Table 2. Among them, *k*_1_, *k*_2_, and *k*_3_ are the mean of *ρ*, *P*, and *Ps* under a level of a certain factor, respectively. *R*_1_, *R*_2_, and *R*_3_ are the range corresponding to *k*_1_, *k*_2_, and to *k*_3_, respectively. Calculated values of *k* and *R* are listed in Table 3.

As for a lightweight reinforced composite material, specific compressive strength (*P_s_*) is a typical mechanical index for LCGFR-LDUPR samples. Table 3 indicates that effects of three factors on *R*_3_ (the range between *k*_3*max*_ and *k*_3*min*_ for *P_s_*) are the content of long chopped glass fiber (factor C), the content of NH_4_HCO_3_ (factor B), and the curing temperature (factor A) sequentially. Among them, *R*_3_ is mostly influenced by factor C (corresponding to the maximum value of 10.28 MPa/g·cm^3^). Next is factor B, corresponding to the value of 2.49 MPa/g·cm^3^. The minimum influence is factor A, corresponding to the value of 1.09 MPa/g·cm^3^. Therefore, it is deduced that factor C was the critical factor for the specific compressive strength (*P_s_*) of LCGFR-LDUPR composite samples.

In light of these results, the specific compressive strength (*P_s_*) of sample 24 in Table 3 reaches up to the highest value of 52.00 ± 0.74 MPa/g·cm^3^ under conditions of 2.50 phr of NH_4_HCO_3_, 20.00 phr of chopped glass fibers (in length of 15.0 mm), 1.50 phr of MEKP-II together with 0.50 phr of cobalt naphthenate, and at the curing temperature of 58.0 °C. Moreover, the apparent density (*ρ*) is 0.68 ± 0.02 g/cm^3^ and the compressive strength (*P*) is 35.36 ± 0.38 MPa, which are the other two essential parameters in Table 3. The value of *P_s_* or *P* of the LCGFR-LDUPR composite sample was 34.9% or 45.6%, higher than that of chopped glass fiber (in length of 3.0 mm) reinforced low-density unsaturated polyester resin (CGFR-LDUPR) sample [15]. Moreover, the value of *P_s_* of LCGFR-LDUPR composite sample was close to that of the chopped carbon fiber (in length of 6.0 mm) reinforced low-density unsaturated polyester resin (CCFR-LDUPR) composite sample, but the value of *P* was 13.2% higher than that of the CCFR-LDUPR composite sample [16].

### 3.5. Reinforcement of Long Chopped Glass Fiber to Low-Density Unsaturated Polyester Resin in the Presence of Different Initiators

Different from previous studies, three kinds of initiators were applied in this research to explore different initiation mechanisms for unsaturated polyester resin. Based on the above experimental results of coordination of methyl ethyl ketone peroxide (MEKP-II) and cobalt naphthenate, the other two initiation combinations, which were cyclohexanone peroxide (CYPH)/cobalt naphthenate and tert-butyl peroxy-2-ethylhexanoate (TBPO)/cobalt naphthenate, were introduced into the preparation of long chopped glass fiber reinforced low-density unsaturated polyester resin (LCGFR-LDUPR) composite samples, respectively. Reinforcement of long chopped glass fiber to low-density unsaturated polyester resin was detected and discussed in the presence of different initiation combinations.

According to experiments of the gel time of resin glue, the proper ratio of CYPH to cobalt naphthenate was 30 to 1, i.e., 1.50 phr of CYPH and 0.05 phr of cobalt naphthenate. Besides, the proper ratio of TBPO to cobalt naphthenate was 15 to 1, i.e., 1.50 phr of TBPO and 0.10 phr of cobalt naphthenate.

In accordance with those conditions, long chopped glass fibers at a length of 15.0 mm and gotten from ER4800-T718 plied yarn were used to prepare LCGFR-LDUPR composite samples through orthogonal experiments of L_25_(5^3^) with same factors and same levels (as same as in Table 2) for two different initiator/accelerator cooperation. One orthogonal experiment was in the presence of 1.50 phr of CYHP and 0.05 phr of cobalt naphthenate, corresponding to results of 58.0 °C curing temperature, 2.50 phr of NH_4_HCO_3_, and 20.00 phr of long chopped glass fibers. The LCGFR-LDUPR composite sample presented the apparent density (*ρ*) of 0.71 ± 0.03 g/cm^3^, the compression strength (*P*) of 35.15 ± 0.40 MPa, and the highest specific compressive strength (*P_s_*) of 49.51 ± 0.65 MPa/g·cm^3^. Another orthogonal experiment was in the presence of 1.50 phr of TBPO and 0.10 phr of cobalt naphthenate, also corresponding to 58.0 °C curing temperature, 2.50 phr of NH_4_HCO_3_, and 20.00 phr of long chopped glass fibers. The apparent density (*ρ*) was 0.63 ± 0.02 g/cm^3^, the compressive strength (*P*) was 28.33 ± 0.32 MPa, and the highest specific compressive strength (*P_s_*) was 44.97 ± 0.54 MPa/g·cm^3^ for the LCGFR-LDUPR composite sample.

It is unambiguous that the initiator/accelerator cooperation of MEKP-II and cobalt naphthenate was the most effective one among the three different initiator/accelerator systems for the preparation of the LCGFR-LDUPR composite sample.

### 3.6. Curing of Unsaturated Polyester Resin (UPR) with Different Initiators and the Same Accelerator

Discussions in paragraph 3.5 show that the specific compressive strength of the long chopped glass fiber reinforced low-density unsaturated polyester resin (LCGFR-LDUPR) composite sample in the presence of methyl ethyl ketone peroxide (MEKP-II)-cobalt naphthenate is the highest. It prompts us to elucidate mechanisms among three different initiators with the presence of cobalt naphthenate. Explorations were made through non-isothermal differential scanning (DSC) and the results are illustrated in Figure 6.

Different initial temperatures, different peak temperatures, different peak widths, and different curing exothermic enthalpies of curing process for the different samples, which are obtained from Figure 6, are listed in Table 4 and Table 5, respectively.

It is analyzed that due to the “bridging” action between the long chopped glass fiber and the linear polymer molecule, which was caused by the coupling agent, the chopped glass fiber led the linear molecular of polyester to close nearby. It was a dominant tendency for the linear molecular crosslinking of polyester. Therefore, the initial temperature of the crosslinking of unsaturated polyester resin is 104.2 °C, a little lower than that of the pure unsaturated polyester resin (105.91 °C) which is shown in Table 4. In Figure 6, curing exothermic curve (b), curing exothermic curve (f), and curing exothermic curve (j) illustrate that NH_4_HCO_3_ (its content high to 2.50 phr) decomposes and releases aqua before crosslinking of unsaturated polyester resin. This decomposition was up to the highest speed in the temperature range of 70 °C to 85 °C [15,17]. Aqua coming from the decomposition of NH_4_HCO_3_ retarded the crosslinking of unsaturated polyester resin and led to a higher initial temperature of crosslinking of unsaturated polyester resin which is 106.94 °C in Table 4. Long chopped glass fibers and NH_4_HCO_3_ exerted synthetic influences on the initial temperature of crosslinking of unsaturated polyester resin for composite samples with the presence of long chopped glass fiber and NH_4_HCO_3_, which are shown in Table 4.

Because of steric retardances of bubbles and long chopped glass fibers, the exothermic peak temperature of unsaturated polyester resin coexisting with foaming agent and long chopped glass fibers is a little higher than that of pure unsaturated polyester resin. Meanwhile, the peak width is also widened.

In the presence of foaming agent and long chopped glass fiber, the percentage of unsaturated polyester resin was relatively decreased, resulting in a decrease in exothermic enthalpy of crosslinking of unsaturated polyester resin coexisting with foaming agent and long chopped glass fibers, which is illustrated in Table 5. In Table 5, it is unambiguous that with coexistence of MEKP-II and cobalt naphthenate, the exothermic enthalpy of crosslinking of unsaturated polyester resin is as low as 187.1 J/g, and the peak width of 48.7 °C is the widest one which represented the slowest speed of crosslinking of unsaturated polyester resin in the sample preparation. It is favorable for the formation, diffusion, and distribution of bubbles, and also for a complete crosslinking of unsaturated polyester resin. It is also an improvement for mechanical properties of LCGFR-LDUPR composite samples. Therefore, thermal analytic results reveal that MEKP-II and cobalt naphthenate performed optimal initiation for the crosslinking of unsaturated polyester resin among three different initiators and the same accelerator.

### 3.7. Initiation Mechanisms of Different Initiators

Experimental results in paragraph 3.6 (Curing of unsaturated polyester resin with different initiators and the same accelerator) reveal that the curing process initiated by methyl ethyl ketone peroxide (MEKP-II) and cobalt naphthenate presented characteristics of low curing exothermic enthalpy and slow curing speed. It is necessary to bring out different mechanisms for different initiators in the curing process of unsaturated polyester resin. Figure 7 indicates the decomposition of cobalt naphthenate, spontaneous formation of radicals for different initiators, and different radical formation processes for different initiators with the presence of cobalt naphthenate [32,33].

There are two moles radical *b* generating with the spontaneous decomposition of MEKP-II shown in Figure 7b. The electron cloud was repelled to move toward the oxygen radical by alkyl groups (–CH_3_ and –C_2_H_5_). The electron cloud on the oxygen radical was increased, and the activity of the oxygen radical was promoted. Two kinds of free radicals (radical *c* and radical *d*) are produced with the reaction between MEKP-Ⅱ and cobalt naphthenate, which are illustrated in Figure 7c,d, respectively. As for radical *c* and radical *d*, electron clouds were repelled to move toward those two oxygen radicals, and their activity were both increased.

Cyclohexanone peroxide (CYHP) releases radical *e*_1_ and radical *e*_2_, two kinds of radicals, as shown in Figure 7e. Electron clouds on radicals were reduced by electron withdrawing groups which were hydroxyl group (–OH) and hydroperoxyl group (–OOH), resulting in a decrease of the activities of radicals. Two kinds of radicals, radical *f* and radical *g*, which are produced in the reaction of CYHP and cobalt naphthenate, are indicated in Figure 7f,g, respectively. However, activities of radical *f* and radical *g* were decreased by adjacent electron-withdrawing groups which were hydroxyl (–OH) and atoms of oxygen.

Tert-butyl peroxy-2-ethylhexanoate (TBPO) releases radical *h*_1_ and radical *h*_2_, two kinds of radicals, as shown in Figure 7h. As for radical *h*_1_, the electron cloud was repelled to move toward the oxygen radical by the methyl group (–CH_3_), resulting in an increase of the electron cloud of radical *h*_1_. On the other hand, the electron cloud was attracted by adjacent carbonyl despite the electron repellency of alkyl groups, resulting in no electron cloud change for radical *h*_2_. Radical *i* was produced in the reaction between TBPO and cobalt naphthenate (see Figure 7i). Because of the existence of three methyl (–CH_3_) groups, the electron cloud was repelled toward radical *i*, resulting in the promotion of the activity of radical *i*.

Initiations of different initiators were not only closely related to activities of radicals, but also related to contents of initiators. The higher dosage of the radical, the higher efficiency of initiation for the radical, and the more accelerated crosslinking for the unsaturated polyester resin [34,35,36]. In the study, contents of MEKP-II, CYHP, and TBPO in their solution were 33 wt%, 50 wt%, and 98 wt%, respectively. According to those corresponding contents and under the condition of 1.50 phr of initiator together with the presence of cobalt naphthenate in resin glue, dosages of radicals could be calculated. As for MEKP-II, 0.12 phr of radical *c* and 0.25 phr of radical *d* were produced, and with a total dosage of radicals of 0.37 phr. Similarly, 0.22 phr of radical *f* and 0.37 phr of radical *g* were produced by CYHP, with a total dosage of radicals of 0.59 phr. As for TBPO, only radical *i* was obtained in Figure 7i, with a total dosage of radical *i* of 0.48 phr. Curing degrees (α) of unsaturated polyester resin with different initiators and the same accelerator were detected and listed in Table 6.

There are two kinds of activated radicals for the initiation of unsaturated polyester resin with the coexistence of MEKP-II and cobalt naphthenate in Figure 7c,d. However, the lowest dosage of radicals might confine the crosslinking of unsaturated polyester resin. Under influences of the activity and the lowest dosage of radicals, the curing degree (α) of resin or the specific compressive strength of the LCGFR-LDUPR sample reached the highest value of 0.75 or 52.00 ± 0.74 MPa/g·cm^3^.

Although the total dosage of radicals was the highest among those three kinds of initiators. Activities of two kinds of radicals, which were produced by the coexistence of CYHP and cobalt naphthenate, were reduced. The synergistic effect of the activity and the dosage of radicals made the curing degree (α) be 0.71, and the specific compressive strength decreased to 49.51 ± 0.65 MPa/g·cm^3^ for the LCGFR-LDUPR sample.

There was only one kind of activated radical *i* produced by the coexistence of TBPO and cobalt naphthenate. Both the increase of activity and the higher dosage of radical *i* were favorable for the acceleration of crosslinking of unsaturated polyester resin. Under the influences of those conditions, the fast polymerization of resin was realized but was not completed. Therefore, the curing degree (α) was 0.64, the lowest one among the cured LCGFR-LDUPR samples with three different kinds of initiators, leading to the lowest specific compressive strength of 44.97 ± 0.54 MPa/g·cm^3^ for the sample.

### 3.8. Microstructure Analyses of Cured Specimens

Distributions of bubbles and long chopped glass fibers in cured low-density unsaturated polyester resin or in long chopped glass fiber reinforced low-density unsaturated polyester resin (LCGFR-LDUPR) composite samples were observed by auto-fine-tune portable microscope (B011), and are shown in Figure 8.

Comparing Figure 8a,b, it is indicated that with the presence of 20.00 phr of long chopped glass fibers and the coexistence of methyl ethyl ketone peroxide (MEKP-II) and cobalt naphthenate, the diameters of bubbles change from 350–460 μm for cured low-density unsaturated polyester resin to 200–250 μm for LCGFR-LDUPR composite sample. In the same way, diameters of bubbles change from 330–450 μm for cured low-density unsaturated polyester resin to 150–250 μm for LCGFR-LDUPR composite sample with the presence of 20.00 phr of long chopped glass fibers and the coexistence of cyclohexanone peroxide (CYHP) and cobalt naphthenate. Moreover, diameters of bubbles change from 250–500 μm for cured low-density unsaturated polyester resin to 180–250 μm for LCGFR-LDUPR composite sample with the presence of 20.00 phr of long chopped glass fibers and the coexistence of tert-butyl peroxy-2-ethylhexanoate (TBPO) and cobalt naphthenate. It is because that long chopped glass fiber occupied the internal volume of samples (see Figure 8c,f,i) and the distribution volume of bubbles was lost. The smaller space for bubbles formation within the unit volume of samples, the smaller diameter for bubbles. Furthermore, the density of LCGFR-LDUPR composite samples increases with the addition of long chopped glass fibers, which is shown in Table 2.

As for cured low-density unsaturated polyester resin with the coexistence of methyl MEKP-II and cobalt naphthenate or the coexistence of CYHP and cobalt naphthenate, bubbles are uniform in size and are shown in Figure 8a,d. However, Figure 8g indicates that with the coexistence of TBPO and cobalt naphthenate, bubbles are nonuniform in size, accompanying the appearance of a few linked bubbles for cured low-density unsaturated polyester resin sample.

As for cured long chopped glass fiber reinforced low-density unsaturated polyester resin (LCGFR-LDUPR) composite samples with the coexistence of different initiators and the same accelerator (see Figure 8b,e,h), bubbles are uniform in size, distribute homogeneously and without linked bubbles only for the cured sample in the presence of MEKP-II and cobalt naphthenate.

Considered together with paragraph 3.6 (Curing of unsaturated polyester resin with different initiators and a same accelerator), it is revealed that with the presence of 2.50 phr of NH_4_HCO_3_, the curing exothermic peak width of unsaturated polyester resin is 46.25 °C, 42.96 °C, and 28.51 °C for the MEKP-II-cobalt naphthenate system, CYHP-cobalt naphthenate system, and TBPO-cobalt naphthenate system, respectively. The curing process of unsaturated polyester resin was evidently slower for the existence of MEKP-II and cobalt naphthenate or CYHP and cobalt naphthenate than of that for the existence of TBPO and cobalt naphthenate. A slower curing process of unsaturated polyester resin was more favorable to generation and distribution of bubbles in composite samples. In the same way, with the presence of 20.00 phr of long chopped glass fibers (in length of 15.0 mm) and 2.50 phr of NH_4_HCO_3_, curing exothermic peak width of unsaturated polyester resin is 48.70 °C, 44.04 °C, and 31.05 °C for MEKP-II-cobalt naphthenate system, CYHP-cobalt naphthenate system, and for the TBPO-cobalt naphthenate system, respectively. It is obviously that MEKP-II and cobalt naphthenate enabled the curing process of unsaturated polyester resin slowest among three different initiation systems, which was most beneficial to generation, diffusion, and distribution of bubbles in composite samples.

Under the condition of MEKP-II, cobalt naphthenate, and 20.00 phr of chopped glass fibers which were in lengths of 3.0 mm, 9.0 mm, 15.0 mm, and 21.0 mm, distributions of bubbles and chopped glass fibers in cured chopped glass fiber reinforced low-density unsaturated polyester resin (CGFR-LDUPR) samples were observed by auto-fine-tune portable microscope (B011), and are shown in Figure 9.

Figure 9a,c indicate that linked bubbles emerge in cured CGFR-LDUPR composite samples which contained 20.00 phr of chopped glass fibers in length of 3.0 mm or 9.0 mm. Figure 9e shows that bubbles are uniform in size and are a homogeneous distribution in the cured long chopped glass fiber reinforced low-density unsaturated polyester resin (LCGFR-LDUPR) sample containing 20.00 phr of chopped glass fibers at a length of 15.0 mm. It is because under the condition of the same addition of chopped glass fiber (such as 20.00 phr), the shorter length of chopped glass fibers was, the more amounts of bundles there were per content of chopped glass fibers. Bundles of chopped glass fibers were toward orientation and cluster in cured CGFR-LDUPR composite samples, which are exhibited in Figure 9b,d. Orientation and cluster of fibers are obvious for the chopped glass fiber at a length of 3.0 mm in Figure 9b. Therefore, bubbles were squeezed by the orientation and the cluster of chopped glass fibers during the preparation of the sample, resulting in adverse effects on the formation and the homogeneous distribution of bubbles, where a number of linked bubbles emerged. However, chopped glass fibers, which were at a length of 15.0 mm, distributed homogeneously and without orientation in the cured sample (see Figure 9f). Moreover, a dimensional framework formed, which was of benefit to the reinforcement of composite samples.

There is deformation and inhomogeneous distribution of bubbles in the cured LCGFR-LDUPR composite specimen (in Figure 9d) with the length of the long chopped glass fibers being 21.0 mm. Chopped glass fibers of 21.0 mm long were easy to warp (see Figure 9h), and made bubbles squeezed, deformed, and unevenly distributed, which is shown in Figure 9g. Schematic diagrams of distributions of bubbles and 20.00 phr of chopped glass fibers (which were in length of 3.0 mm, 9.0 mm, 15.0 mm, and 21.0 mm) in cured samples are illustrated in Figure 10. Deformed and unevenly distributed bubbles, and wrapped long chopped glass fibers, decreased the reinforcement of long chopped glass fibers to composite samples.

With the coexistence of methyl ethyl ketone peroxide (MEKP-II) and cobalt naphthenate, and with the presence of long chopped glass fibers in different contents (such as 10.00 phr, 15.00 phr, 20.00 phr, and 25.00 phr), distributions of bubbles and long chopped glass fibers in cured long chopped glass fiber reinforced low-density unsaturated polyester resin (LCGFR-LDUPR) composite specimens were observed by the auto-fine-tune portable microscope (B011), and are shown in Figure 11. Figure 11a,c,e,g illustrate that diameters of bubbles change from range of 320–400 μm to range of 150–250 μm as the content increase of long chopped glass fibers. With the content of long chopped glass fiber being 25.00 phr for the LCGFR-LDUPR composite specimen, bubbles are different in size and are in a homogeneous distribution. Linked bubbles are also found in the composite specimen. On the other hand, Figure 11b,d,f indicate that corresponding to 10.00 phr, 15.00 phr, and 20.00 phr long chopped glass fibers, all of chopped glass fibers distribute homogeneously in cured LCGFR-LDUPR composite specimens. However, long chopped glass fibers have serious orientation in the cured LCGFR-LDUPR composite sample with the presence of 25.00 phr of long chopped glass fibers, which is shown in Figure 11h.

Schematic diagrams of distributions of bubbles and chopped glass fibers (which were at a length of 15.0 mm, and in the presence of 10.00 phr, 15.00 phr, and 25.00 phr) in cured samples are illustrated in Figure 12. Comparing Figure 10c and Figure 12a,b, it is revealed that in Figure 10c, long chopped glass fibers distribute most homogeneously, which indicates that the long chopped glass fiber reinforced low-density unsaturated polyester resin (LCGFR-LDUPR) composite sample with the presence of 20.00 phr of long chopped glass fibers presented the best mechanical performances among those three samples. Figure 12c describes that the long chopped glass fibers in the presence of 25.00 phr have serious orientation in the cured sample. Those phenomena emerge in Figure 11h. Fibers serious orientation prevented the growth of bubbles, accompanying a number of linked bubbles. Furthermore, fibers serious orientation was adverse to the reinforcement of long chopped glass fibers to composite material.

## 4. Conclusions

Long chopped glass fibers reinforced low-density unsaturated polyester resin (LCGFR-LDUPR) composite samples with light weight and excellent mechanical properties were prepared in this study. Long chopped glass fiber, which was in length of 15.0 mm and chopped from ER4800-T718 plied yarn, was the proper reinforce material in the application. By the initiation of 1.50 phr of methyl ethyl ketone peroxide (MEKP-II) and 0.05 phr of cobalt naphthenate, 20.00 phr of long chopped glass fiber and 2.50 phr of NH_4_HCO_3_ were applied to prepare LCGFR-LDUPR samples at a temperature of 58.0 °C through orthogonal experiments. The LCGFR-LDUPR composite sample exhibited the density (*ρ*) of 0.68 ± 0.02 g/cm^3^, the compression strength (*P*) of 35.36 ± 0.38 MPa, and the highest specific compressive strength (*P_s_*) of 52.00 ± 0.74 MPa/g·cm^3^. The value of *P_s_* of the LCGFR-LDUPR sample was much higher than that of chopped glass fiber (in length of 3.0 mm) reinforced low-density unsaturated polyester resin (CGFR-LDUPR) which was in the value of 38.56 ± 0.62 MPa/g·cm^3^. Furthermore, the value of *P_s_* of the LCGFR-LDUPR composite sample was close to that of a chopped carbon fiber (at a length of 6.0 mm) reinforced low-density unsaturated polyester resin (CCFR-LDUPR) composite sample, which was in the value of 53.56 ± 0.83 MPa/g·cm^3^.

Microscopic analyses show that long chopped glass fibers constituted the ideal three-dimensional framework of fibers in steric of samples. Bubbles were uniform in size and distributed homogenously in the framework of fibers without bubbles deformation and linked bubbles.

Novel initiation mechanisms were discussed and presented in the work. It was pointed out that radicals activated by an increase of electron clouds on radicals were favorable for the crosslinking of unsaturated polyester resin. Low dosage of radicals slowed down the speed of crosslinking of resin, which was for methyl ethyl ketone peroxide (MEKP-II) or for cyclohexanone peroxide (CYHP). The crosslinking of unsaturated polyester resin was influenced by both the activity changes and the dosage of radicals. Mechanism studies of three different initiators with the same accelerator illustrated that methyl ethyl ketone peroxide (MEKP-II) and cobalt naphthenate produced activated radicals. The total dosage of those activated radicals was the lowest one compared with that of radicals produced by cyclohexanone peroxide (CYHP) and cobalt naphthenate or tert-butyl peroxy-2-ethylhexanoate (TBPO) and cobalt naphthenate. The lowest dosage of activated radicals produced by MEKP-II and cobalt naphthenate enabled the lower curing exothermic enthalpy and the slowest crosslinking for unsaturated polyester resin to be carried out, resulting in complete curing and a higher curing degree of resin. That lower curing exothermic enthalpy and the slowest crosslinking of resin were conducive to the formation, diffusion, and distribution of bubbles in uniform size, and also for the constitution of ideal three-dimensional framework. Homogenous distribution of bubbles, the optimal dimensional framework of long chopped glass fiber in length of 15.0 mm, lower curing exothermic enthalpy, and the slowest crosslinking of resin promoted excellent mechanical properties of the LCGFR-LDUPR composite sample.

## Figures and Tables

**Figure 1 materials-14-07307-f001:**
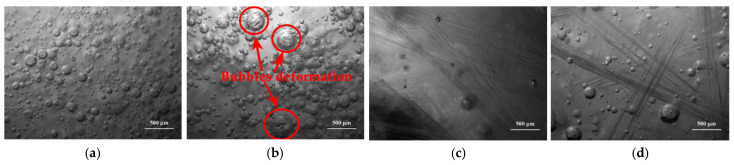
Micrographs of distributions of different kinds of chopped glass fibers, in length of 9.0 mm and with content of 10.00 phr, in resin glue (**a**) in type ECR13-300D-608, (**b**) in type ECR15-600D-608, (**c**) in type ER2400-T628, (**d**) in type ER4800-T718.

**Figure 2 materials-14-07307-f002:**
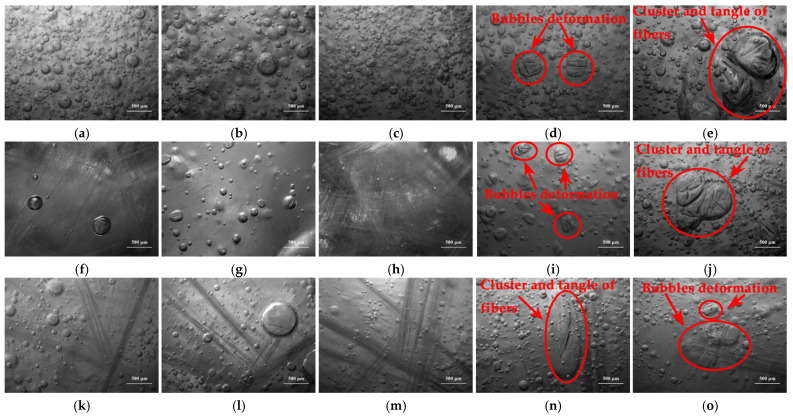
Micrographs of distributions of 15.00 phr of chopped glass fibers in resin glue (**a**) at a length of 9.0 mm, (**b**) at a length of 12.0 mm, (**c**) at a length of 15.0 mm, (**d**) at a length of 18.0 mm, (**e**) at a length of 21.0 mm got from type ECR13-300D-608, (**f**) at a length of 9.0 mm, (**g**) at a length of 12.0 mm, (**h**) at a length of 15.0 mm, (**i**) at a length of 18.0 mm, (**j**) at a length of 21.0 mm got from type ER2400-T628, (**k**) at a length of 9.0 mm, (**l**) at a length of 12.0 mm, (**m**) at a length of 15.0 mm, (**n**) at a length of 18.0 mm, (**o**) at a length of 21.0 mm got from type ER4800-T718.

**Figure 3 materials-14-07307-f003:**
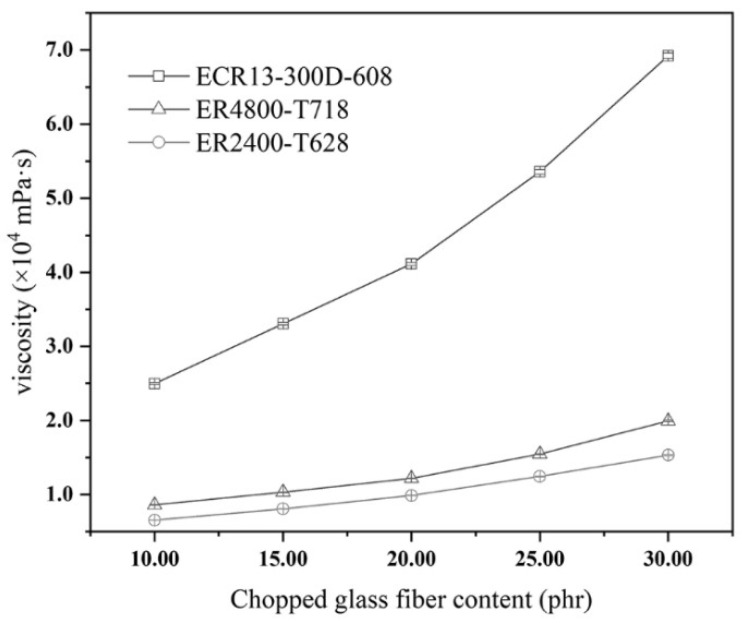
Effects of three types of chopped glass fiber at a length of 15.0 mm and in different contents (10.00 phr, 15.00 phr, 20.00 phr, 25.00 phr, and 30.00 phr) on the viscosity of the resin glue.

**Figure 4 materials-14-07307-f004:**
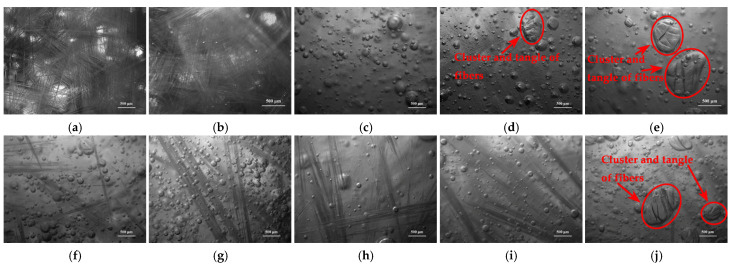
Micrographs of distributions of 15.0 mm chopped glass fibers in resin glue: (**a**) in the presence of 10.00 phr, (**b**) in the presence of 15.00 phr, (**c**) in the presence of 20.00 phr, (**d**) in the presence of 25.00 phr, (**e**) in the presence of 30.00 phr got from type ER2400-T628, (**f**) in the presence of 10.00 phr, (**g**) in the presence of 15.00 phr, (**h**) in the presence of 20.00 phr, (**i**) in the presence of 25.00 phr, (**j**) in the presence of 30.00 phr got from type ER4800-T718.

**Figure 5 materials-14-07307-f005:**
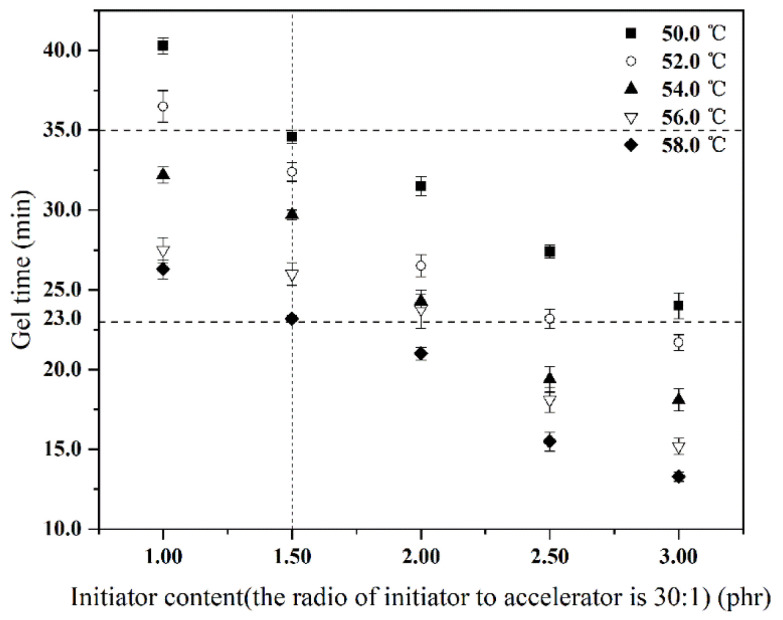
Gel time of resin glue with different contents of initiator MEKP-II and accelerator at temperature from 50.0 °C to 58.0 °C.

**Figure 6 materials-14-07307-f006:**
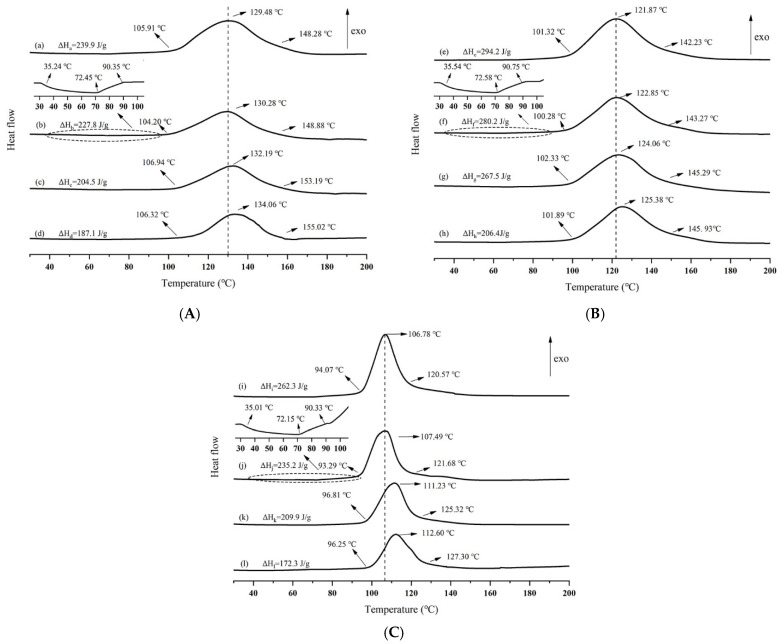
Non-isothermal Differential Scanning Calorimetry curves of (**A**) wherein is (**a**) pure UPR, (**b**) UPR + 20.00 phr of long chopped glass fiber, (**c**) UPR + 2.50 phr of NH_4_HCO_3_, (**d**) UPR + 20.00 phr of long chopped glass fiber + 2.50 phr of NH_4_HCO_3_, in presence of MEKP-II and cobalt naphthenate; (**B**) wherein is (**e**) pure UPR, (**f**) UPR + 20.00 phr of long chopped glass fiber, (**g**) UPR + 2.50 phr of NH_4_HCO_3_, (**h**) UPR + 20.00 phr of long chopped glass fiber + 2.50 phr of NH_4_HCO_3_, in presence of CYHP and cobalt naphthenate; (**C**) wherein is (**i**) pure UPR, (**j**) UPR + 20.00 of phr long chopped glass fiber, (**k**) UPR + 2.50 phr of NH_4_HCO_3_, (**l**) UPR + 20.00 phr of long chopped glass fiber + 2.50 phr of NH_4_HCO_3_, in presence of TBPO and cobalt naphthenate.

**Figure 7 materials-14-07307-f007:**
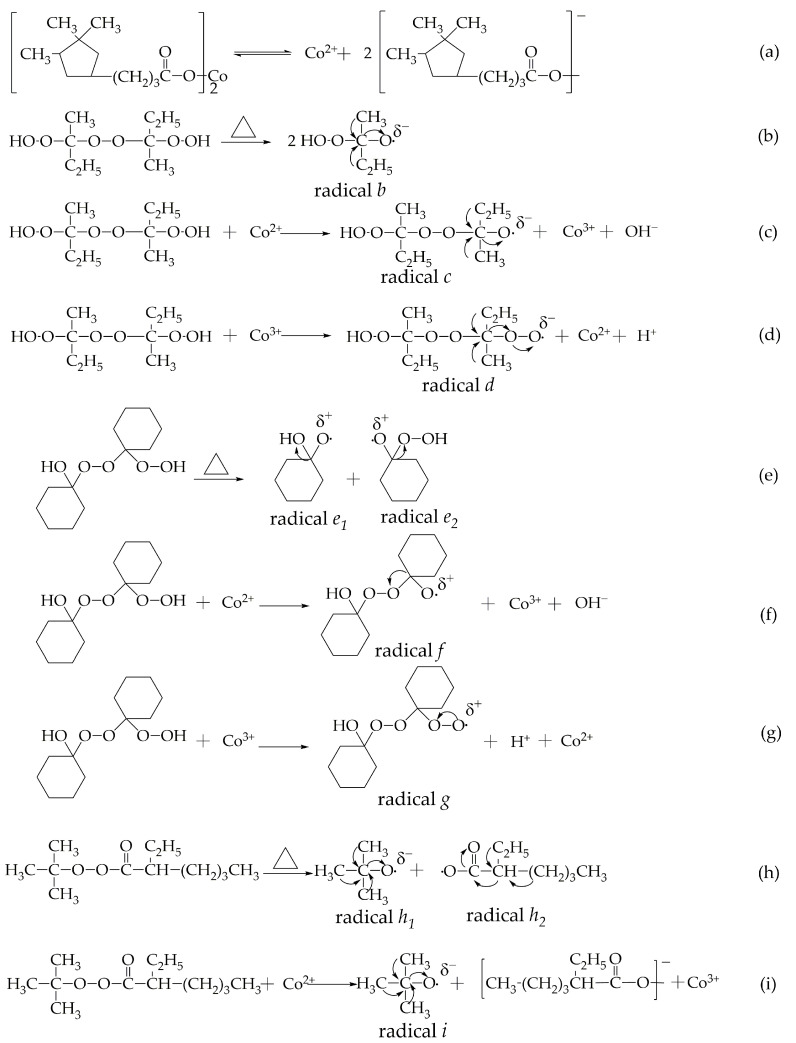
Decomposition equations of accelerator and initiators, (**a**) decomposition of accelerator cobalt naphthenate, (**b**) spontaneous decomposition of MEKP-Ⅱ, (**c**,**d**) decomposition of MEKP-Ⅱ with the promotion of cobalt ions, (**e**) spontaneous decomposition of CYHP, (**f**,**g**) decomposition of CYHP with the promotion of cobalt ions, (**h**) spontaneous decomposition of MEKP-Ⅱ, (**i**) decomposition of TBPO with the promotion of cobalt ions.

**Figure 8 materials-14-07307-f008:**
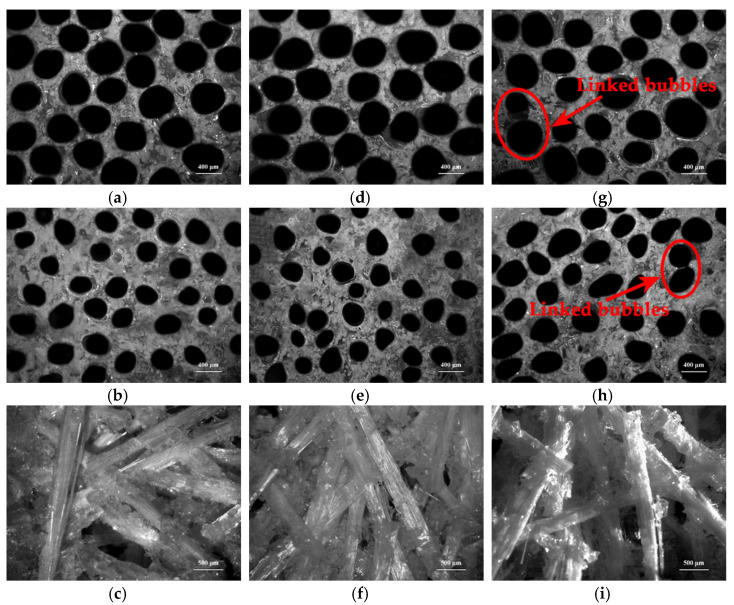
Micrographs of cured samples, with the coexistence of MEKP-II and cobalt naphthenate, (**a**) cured LDUPR sample, (**b**,**c**) cured LCGFR-LDUPR composite sample with the presence of 20.00 phr of long chopped glass fiber; with the coexistence of CYHP and cobalt naphthenate, (**d**) cured LDUPR sample, (**e**,**f**) cured LCGFR-LDUPR composite sample with the presence of 20.00 phr of long chopped glass fiber; with the coexistence of TBPO and cobalt naphthenate, (**g**) cured LDUPR sample, (**h**,**i**) cured LCGFR-LDUPR composite sample with the presence of 20.00 phr of long chopped glass fiber.

**Figure 9 materials-14-07307-f009:**
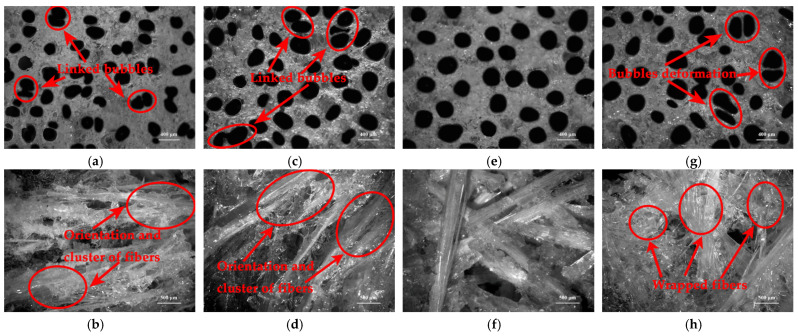
Micrographs of cured samples with the presence of 20.00 phr long chopped glass fiber and the coexistence of MEKP-II and cobalt naphthenate: (**a**,**b**) cured CGFR-LDUPR composite sample in the presence of 3.0 mm chopped glass fiber, (**c**,**d**) cured CGFR-LDUPR composite sample in the presence of 9.0 mm chopped glass fiber, (**e**,**f**) cured LCGFR-LDUPR composite sample in the presence of 15.0 mm chopped glass fiber, (**g**,**h**) cured LCGFR-LDUPR composite sample in the presence of 21.0 mm chopped glass fiber.

**Figure 10 materials-14-07307-f010:**
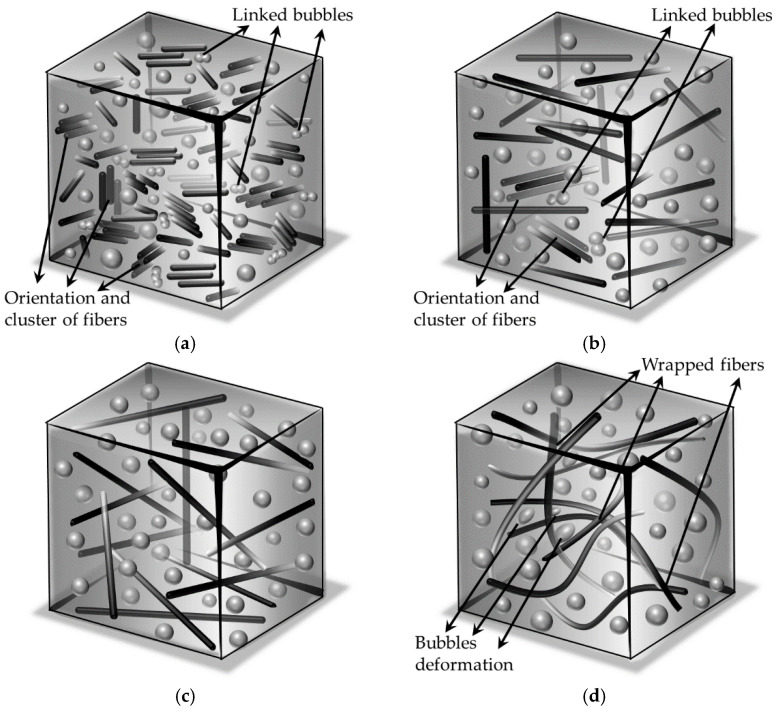
Schematic diagrams of distributions of bubbles and 20.00 phr of chopped glass fiber: (**a**) at a length of 3.0 mm, (**b**) at a length of 9.0 mm, (**c**) at a length of 15.0 mm, (**d**) at a length of 21.0 mm in cured samples.

**Figure 11 materials-14-07307-f011:**
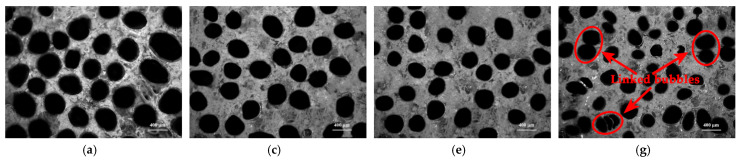
Micrographs of cured samples with the presence of 15.0 mm long chopped glass fiber, with the coexistence of MEKP-II and cobalt naphthenate: (**a**,**b**) cured LCGFR-LDUPR composite specimen with the presence of 10.00 phr of long chopped glass fiber, (**c**,**d**) cured LCGFR-LDUPR composite specimen with the presence of 15.00 phr of long chopped glass fiber, (**e**,**f**) cured LCGFR-LDUPR composite specimen with the presence of 20.00 phr of long chopped glass fiber, (**g**,**h**) cured LCGFR-LDUPR composite specimen with the presence of 25.00 phr of long chopped glass fiber.

**Figure 12 materials-14-07307-f012:**
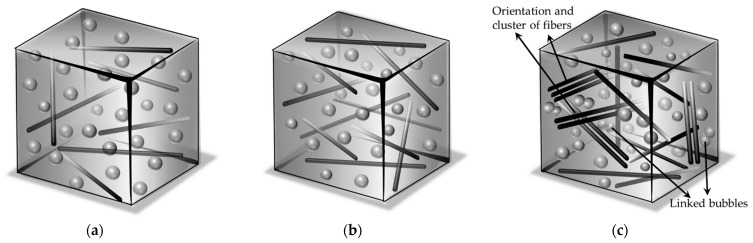
Schematic diagrams of distributions of bubbles and 15.0 mm chopped glass fiber: (**a**) with the presence of 10.00 phr, (**b**) with the presence of 15.00 phr, (**c**) with the presence of 25.00 phr in cured samples.

**Table 1 materials-14-07307-t001:** Gel time of resin glue in different ratios of initiator to accelerator.

Ratio of Initiator to Accelerator	10:1	20:1	30:1	40:1	50:1
Gel time of resin at 50.0 °C (min)	9.7 ± 0.4	20.5 ± 0.3	31.5 ± 0.6	42.7 ± 0.5	55.1 ± 0.7
Gel time of resin at 58.0 °C (min)	4.3 ± 0.2	9.5 ± 0.3	21.0 ± 0.3	35.9 ± 0.4	51.2 ± 0.6

**Table 2 materials-14-07307-t002:** Orthogonal experiment design and results of LCGFR-LDUPR composite samples.

Sample Serial Number	Curing Temperature (°C) (A)	NH_4_HCO_3_ Content (phr) (B)	Long Chopped Glass Fiber Content (phr) (C)	*ρ* (g/cm^3^)	*P* (MPa)	*P_s_* (MPa/g·cm^3^)
1	50.0	1.00	10.00	0.78 ± 0.02	31.89 ± 0.45	40.88 ± 0.55
2	50.0	1.50	15.00	0.81 ± 0.03	31.56 ± 0.36	38.96 ± 0.36
3	50.0	2.00	20.00	0.80 ± 0.03	35.63 ± 0.56	44.54 ± 0.54
4	50.0	2.50	25.00	0.82 ± 0.04	32.45 ± 0.62	39.57 ± 0.44
5	50.0	3.00	30.00	0.88 ± 0.01	33.02 ± 0.38	37.52 ± 0.32
6	52.0	1.00	15.00	0.81 ± 0.02	33.14 ± 0.41	40.91 ± 0.29
7	52.0	1.50	20.00	0.82 ± 0.03	36.21 ± 0.80	44.16 ± 0.68
8	52.0	2.00	25.00	0.80 ± 0.01	35.64 ± 0.73	44.55 ± 0.46
9	52.0	2.50	30.00	0.87 ± 0.01	33.69 ± 0.26	38.72 ± 0.51
10	52.0	3.00	10.00	0.66 ± 0.04	23.56 ± 0.32	35.70 ± 0.38
11	54.0	1.00	20.00	0.84 ± 0.02	36.95 ± 0.33	43.99 ± 0.40
12	54.0	1.50	25.00	0.77 ± 0.03	36.85 ± 0.57	47.86 ± 0.63
13	54.0	2.00	30.00	0.90 ± 0.01	30.59 ± 0.63	33.99 ± 0.56
14	54.0	2.50	10.00	0.65 ± 0.02	25.77 ± 0.22	39.65 ± 0.41
15	54.0	3.00	15.00	0.75 ± 0.02	28.01 ± 0.55	37.35 ± 0.66
16	56.0	1.00	25.00	0.80 ± 0.03	37.56 ± 0.68	46.95 ± 0.75
17	56.0	1.50	30.00	0.85 ± 0.01	30.26 ± 0.39	35.60± 0.36
18	56.0	2.00	10.00	0.75 ± 0.04	26.57 ± 0.46	35.43 ± 0.52
19	56.0	2.50	15.00	0.73 ± 0.02	28.33 ± 0.32	38.81 ± 0.66
20	56.0	3.00	20.00	0.78 ± 0.03	34.26 ± 0.44	43.92 ± 0.79
21	58.0	1.00	30.00	0.93 ± 0.03	29.18 ± 0.28	31.38 ± 0.53
22	58.0	1.50	10.00	0.70 ± 0.01	26.10 ± 0.61	37.29 ± 0.71
23	58.0	2.00	15.00	0.72 ± 0.04	31.45 ± 0.57	43.68 ± 0.86
24	58.0	2.50	20.00	0.68 ± 0.02	35.36 ± 0.38	52.00 ± 0.74
25	58.0	3.00	25.00	0.81 ± 0.04	33.86 ± 0.58	41.80 ± 0.56

**Table 3 materials-14-07307-t003:** Values of *k* and *R* of LCGFR-LDUPR composite samples.

Factors	Mean	Level 1	Level 2	Level 3	Level 4	Level 5	*R*_1_, *R*_2_ or *R*_3_ (*k_max_* − *k_min_*)
Curing temperature (A)	$k_{1}^{A}$ (g/cm^3^)	0.82	0.79	0.78	0.78	0.77	0.05
	$k_{2}^{A}$ (MPa)	32.91	32.45	31.63	31.40	31.19	1.72
	$k_{3}^{A}$ (MPa/g·cm^3^)	40.30	40.81	40.57	40.14	41.23	1.09
Content of NH_4_HCO_3_ (B)	$k_{1}^{B}$ (g/cm^3^)	0.83	0.79	0.79	0.75	0.78	0.08
	$k_{2}^{B}$ (MPa)	33.74	32.20	31.98	31.12	30.54	3.20
	$k_{3}^{B}$ (MPa/g·cm^3^)	40.82	40.77	40.44	41.75	39.26	2.49
Content of chopped carbon fiber (C)	$k_{1}^{c}$ (g/cm^3^)	0.71	0.76	0.78	0.80	0.89	0.18
	$k_{2}^{c}$ (MPa)	26.78	30.50	35.68	35.27	31.35	8.90
	$k_{3}^{c}$ (MPa/g·cm^3^)	37.79	39.94	45.72	44.15	35.44	10.28

*k*_1_: the mean of *ρ* calculated from five values under one level for a single factor, *R_1_*: the range between *k_1max_* and *k_1min_*; *k*_2_: the mean of *P* calculated from five values under one level for a single factor, *R_2_*: the range between *k*_2*max*_ and *k*_2*min*_; *k*_3_: the mean of *P_s_* calculated from five values under one level for a single factor, *R_3_*: the range between *k*_3*max*_ and *k*_3*min*_. Subscripts “A”, “B”, or “C” represent curing temperature (factor A), the content of NH_4_HCO_3_ (factor B), and the content of long chopped glass fiber (factor C), respectively.

**Table 4 materials-14-07307-t004:** Different initial temperatures and different peak temperatures of different samples with the presence of different initiators and the same accelerator.

Sample	Initial Temperature (°C)			Peak Temperature (°C)		
	MEKP-II	CYHP	TBPO	MEKP-II	CYHP	TBPO
UPR	105.91	101.32	94.07	129.48	121.87	106.78
UPR + LCGF	104.20	100.28	93.29	130.28	122.85	107.49
UPR + NH_4_HCO_3_	106.94	102.33	96.81	132.19	124.06	111.23
UPR + NH_4_HCO_3_ + LCGF	106.32	101.89	96.25	134.06	125.38	112.60

**Table 5 materials-14-07307-t005:** Different peak widths and different curing exothermic enthalpies of different samples with the presence of different initiators and the same accelerator.

Sample	Peak Width (°C)			Exothermic Enthalpy (J/g)		
	MEKP-II	CYHP	TBPO	MEKP-II	CYHP	TBPO
UPR	42.37	40.91	26.50	239.9	294.2	262.3
UPR + LCGF	44.68	42.99	28.39	227.8	280.2	235.2
UPR + NH_4_HCO_3_	46.25	42.96	28.51	204.5	267.5	209.9
UPR + NH_4_HCO_3_ + LCGF	48.70	44.04	31.05	187.1	206.4	172.3

**Table 6 materials-14-07307-t006:** Curing degrees of unsaturated polyester resin with presences of different initiators.

Sample	*Q_p_* (J/g)	*Q_R_* (J/g)	*Q_T_* (J/g)	Curing Degree (*α*)
LCGFR-LDUPR sample with MEKP-II and cobalt naphthenate	142.9	47.6	190.5	0.75
LCGFR-LDUPR sample with CYHP and cobalt naphthenate	149.4	61.0	210.4	0.71
LCGFR-LDUPR sample with TBPO and cobalt naphthenate	112.4	63.2	175.6	0.64

## Data Availability

Not applicable.

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
