# Peer review of "Long Chopped Glass Fiber Reinforced Low-Density Unsaturated Polyester Resin under Different Initiation"

_materials, 2021, doi:10.3390/ma14237307_

Round 1
Reviewer 1 Report
Authors present a manuscript on the preparation and characterization of composite materials based on long chopped glass fiber reinforced low-density unsaturated polyester resin.
The introduction they present about glass fibers and unsaturated polyester resins can be improved, as it is quite short and seem that several research works on the field are missing.
Materials employed and experimental characterization techniques used for measuring several properties of composites, together with the preparation procedure for the composite materials, are well explained and presented. However, the way they calculate the curing degree by DSC experiments should be better explained...some sentences are not well understandable (...for an hour to obtain the polymerization calorimeter of unsaturated polyester resin which was used as QP...).
Regarding results and discussion, which is the technique employed for obtaining images of figure 1? Should be pointed out in the caption and also in the text. Images themselves could also be improved. The same for figure 2. Moreover, they write: As shown in Figure 2a, 2b, 2c, 2f, 2g, 2h, 2k, 2l, and Figure 2m.... this should be rewritten in another form. The same can be said for Figure 4.
Regarding "Coordination of methyl ethyl ketone peroxide and cobalt naphthenate" and "Preparation of LCGFR-LDUPR composite samples", there is a huge number of samples that have been studied, it seems that in a correct way.
They also properly show the curing process with different initiators and the same accelerator, with a detailed experimental work correctly performed. I also agree with the mechanism proposed for the decomposition of initiators and accelerator.
Micrographs presented for cured samples show good quality and are properly presented, and well explained with the help os schematic diagrams shown in figure 10 and 12.
Conclusions presented are in agreement with obtained data, and the best parameters for their purpose well defined both at conclusions and abstract.
Author Response
Dear reviewer and editors,
Thank you for your letter and for your instructive comments concerning our submitted manuscript entitled "Long Chopped Glass Fiber Reinforced Low-Density Unsaturated Polyester Resin under Different Initiation" (ID: materials-1462468). We made improvements carefully in the revised manuscript. Corrections in the paper and responses to comments of reviewer and editors are listed below:
Response to Reviewer 1:
Comment 1: The introduction they present about glass fibers and unsaturated polyester resins can be improved, as it is quite short and seem that several research works on the field are missing.
Response: Thank you for your suggestion. Unsaturated polyester resin is one of the most wildly used thermosetting resin due to its low cost and excellent processing performances [1-3]. Glass fiber reinforced unsaturated polyester resin composite materials are the most wildly commercial fiber reinforced composite materials for their excellent anti-corrosion and high mechanical properties [4-6]. These introductions have been added in the revised manuscript.
References
- Penczek, P.; Czub, P.; Pielichowski, J., Unsaturated polyester resins: Chemistry and technology. In Crosslinking in Materials Science, Ameduri, B., Ed. 2005; Vol. 184, pp 1-95.
- Chu, F. K.; Hou, Y. B.; Liu, L. X.; Qiu, S. L.; Cai, W.; Xu, Z. M.; Song, L.; Hu, W. Z., Hierarchical Structure: An effective Strategy to Enhance the Mechanical Performance and Fire Safety of Unsaturated Polyester Resin. Acs Applied Materials & Interfaces 2019, 11, (32), 29436-29447.
- Liu, L. B.; Xu, Y.; Xu, M. J.; He, Y. T.; Li, S.; Li, B., An efficient synergistic system for simultaneously enhancing the fire retardancy, moisture resistance and electrical insulation performance of unsaturated polyester resins. Materials & Design 2020, 187.
- Schutte, C. L., ENVIRONMENTAL DURABILITY OF GLASS-FIBER COMPOSITES. Materials Science & Engineering R-Reports 1994, 13, (7), 265-323.
- Sapuan, S. M.; Aulia, H. S.; Ilyas, R. A.; Atiqah, A.; Dele-Afolabi, T. T.; Nurazzi, M. N.; Supian, A. B. M.; Atikah, M. S. N., Mechanical Properties of Longitudinal Basalt/Woven-Glass-Fiber-reinforced Unsaturated Polyester-Resin Hybrid Composites. Polymers 2020, 12, (10).
- Chea, C. P.; Bai, Y.; Fang, Y. H.; Zhang, Y. M., Geometric forming and mechanical performance of reciprocal frame structures assembled using fibre reinforced composites. Engineering Structures 2022, 250.
Comment 2: Materials employed and experimental characterization techniques used for measuring several properties of composites, together with the preparation procedure for the composite materials, are well explained and presented. However, the way they calculate the curing degree by DSC experiments should be better explained...some sentences are not well understandable (...for an hour to obtain the polymerization calorimeter of unsaturated polyester resin which was used as QP...).
Response: Thank you. In the paragraph 2.2.7.2., It is dealt out that value of QT, can be obtained by formula (1). Values of QP and QR were obtained by isothermal differential scanning calorimetry (DSC) experiments according to reference 25. The expression of “the polymerization calorimeter” is wrong, it should be expressed as “the polymerization heat”. Those have been rewritten in the revised manuscript.
References
- Hong, C. M.; Wang, X. J.; Pan, Z. G.; Zhang, Y. F., Curing thermodynamics and kinetics of unsaturated polyester resin with different chain length of saturated aliphatic binary carboxylic acid. Journal of Thermal Analysis and Calorimetry 2015, 122, (1), 427-436.
Comment 3: Regarding results and discussion, which is the technique employed for obtaining images of figure 1? Should be pointed out in the caption and also in the text. Images themselves could also be improved. The same for figure 2. Moreover, they write: As shown in Figure 2a, 2b, 2c, 2f, 2g, 2h, 2k, 2l, and Figure 2m.... this should be rewritten in another form. The same can be said for Figure 4.
Response: Thank you for your advice. The technique employed for obtaining images of figure 1 was indicated in the paragraph of 2.2.1. It is also pointed out in the revised manuscript.
Expressions of Figure 2a, 2b, 2c, etc. have been revised Figure 2a-c, and so on. It is also for Figure 4 revision.
We would like to appreciate your kind instructions on our paper again and we look forward to your response regarding the revised manuscript.
Sincerely yours, Prof. & Dr. Xiaojun Wang
xjwang@njtech.edu.cn
Reviewer 2 Report
This article deals with Glass Fiber Reinforced composite material. After the first reading I did not found a specific scientific contribution, even when it is full of images, data and explanation. In the sencond reading of this extensive paper (23 pages), I realized that in the abstract it is not mendioned the purpose of the paper, how to iniciate the formation? the internal structure? (curing degree, distribution of bublles,the mechanical properties?). The reader may feel a bit confused about the procedure (explained on page 2 lines 75-83) and it is difficult to follow and find the conexion of the methods used (pages 3 and 4). Although there is interesting data and experimental work done, the way of explain and present them must be improved and present a new insight of this type of composites.
Author Response
Dear reviewer and editors,
Thank you for your letter and for your instructive comments concerning our submitted manuscript entitled "Long Chopped Glass Fiber Reinforced Low-Density Unsaturated Polyester Resin under Different Initiation" (ID: materials-1462468). We made improvements carefully in the revised manuscript. Corrections in the paper and responses to comments of reviewer and editors are listed below:
Response to Reviewer 2:
Comment 1: After the first reading I did not find a specific scientific contribution, even when it is full of images, data and explanation.
Response: Thank you for your instruction. Two aspects are included in the manuscript and were described in the title of the manuscript. One is that long chopped glass fiber was applied in low-density unsaturated polyester resin, which the length of 15.0 mm was much longer than the common length of chopped glass fiber. Another is that the exploration of proper initiation mechanism for the sample preparation did not report in previous literatures. In other words, the work carried out an ideal three-dimensional framework of long chopped glass fiber in the reinforcement to low-density unsaturated polyester resin composite samples. It also presented the proper initiator/accelerator of the lower curing exothermic enthalpy and the slowest crosslinking for unsaturated polyester resin. We have followed your instruction to modify the “Introduction” in the revised manuscript.
Comment 2: In the second reading of this extensive paper (23 pages), I realized that in the abstract it is not mentioned the purpose of the paper, how to initiate the formation? the internal structure? (curing degree, distribution of bubbles, the mechanical properties?).
Response: Thank you. According to response comment 1, the purpose of the paper, the initiation, and the internal structure specialities are briefly introduced into the “Abstract” of the revised manuscript.
Comment 3: The reader may feel a bit confused about the procedure (explained on page 2 lines 75-83) and it is difficult to follow and find the connexion of the methods used (pages 3 and 4))
Response: Yes, that is a good idea. The logic of the study and the connexion of the methods used have been provided in the “Introduction” of the revised manuscript.
Comment 4: Although there is interesting data and experimental work done, the way of explain and present them must be improved and present a new insight of this type of composites.
Response: Thanks. We have followed your instruction to present new insights of this type of composites in the “Abstract” and in the “Introduction”.
We would like to appreciate your kind instructions on our paper again and we look forward to your response regarding the revised manuscript.
Sincerely yours,
Prof. & Dr. Xiaojun Wang
xjwang@njtech.edu.cn
Reviewer 3 Report
The author prepared a Long chopped glass fiber reinforced low-density unsaturated polyester resin composite materials. They investigated three Initiation mechanisms using three kinds of initiators with the same accelerator to the crosslinking of unsaturated polyester resin. Overall, the research idea is very innovative and the quality of the presentation is very high. However, I have only two points for author consideration.
1- In Figure 1 (b) I do not see bubbles deformation clearly. could you magnify, please? Also, it was indicated that in figure 1 D "{the bubbles and chopped glass fibers distribute homogeneously in resin glue"; however, I do not see uniform distribution. explain, please.
2- In Figure 2, I noticed many arrows pointing to clusters and tangles of fibers, but they are not clear to me. I suggest viewing those defects by looking at the cross-section instead of just the surface image
Author Response
Dear reviewer and editors,
Thank you for your letter and for your instructive comments concerning our submitted manuscript entitled "Long Chopped Glass Fiber Reinforced Low-Density Unsaturated Polyester Resin under Different Initiation" (ID: materials-1462468). We made improvements carefully in the revised manuscript. Corrections in the paper and responses to comments of reviewer and editors are listed below:
Response to Reviewer 3:
Comment 1: In Figure 1 (b) I do not see bubbles deformation clearly. Could you magnify, please? Also, it was indicated that in figure 1 D "the bubbles and chopped glass fibers distribute homogeneously in resin glue"; however, I do not see uniform distribution. Explain, please.
Response: Thank you for your suggestions. Bubbles’ deformation does not be clearly pointed out in Figure 1. In the revised manuscript, we have marked bubbles’ deformation in Figure 1.
As for "the bubbles and chopped glass fibers distribute homogeneously in resin glue" in paragraph 3.1, it is an improper description. Figure 1 only illustrates the bubbles’ deformation under different conditions. Revisions have been made in the revised manuscript.
Comment 2: In Figure 2, I noticed many arrows pointing to clusters and tangles of fibers, but they are not clear to me. I suggest viewing those defects by looking at the cross-section instead of just the surface image.
Response: Thank you. It is a pity that clusters and tangles of fibers have not been marked clearly in Figure 2. Improvements have been made in the revised manuscript. On the other hand, chopped glass fibers and bubbles, which were in resin glue, were visible. Photos in the manuscript were taken in good depth of field from the side of sample. Differences of chopped glass fibers and bubbles, which were in resin glue, could be shown clearly in micrographs. (please see the diagram below)
We would like to appreciate your kind instructions on our paper again and we look forward to your response regarding the revised manuscript.
Sincerely yours,
Prof. & Dr. Xiaojun Wang
xjwang@njtech.edu.cn
Round 2
Reviewer 1 Report
As authors have now addressed the required changes and comments, the manuscript can be considered as publishable in its actual form
Reviewer 2 Report
The qualiy of the manuscript has improved. I will recomend to be more concise in the abstract and the conclusions